# Robust Allocations with Diversity Constraints

**Zeyu Shen**
Duke University
Durham NC 27708-0129
`zeyu.shen@duke.edu`

**Lodewijk Gelauff**   **Ashish Goel**
Management Science and Engineering
Stanford University, Stanford CA 94305
`{lodewijk,ashishg}@stanford.edu`

**Aleksandra Korolova**
Department of Computer Science
University of Southern California
`korolova@usc.edu`

**Kamesh Munagala**
Department of Computer Science
Duke University, Durham NC 27708-0129
`kamesh@cs.duke.edu`

## Abstract

We consider the problem of allocating divisible items among multiple agents, and consider the setting where any agent is allowed to introduce *diversity constraints* on the items they are allocated. We motivate this via settings where the items themselves correspond to user ad slots or task workers with attributes such as race and gender on which the principal seeks to achieve demographic parity. We consider the following question: When an agent expresses diversity constraints into an allocation rule, is the allocation of other agents hurt significantly? If this happens, the cost of introducing such constraints is disproportionately borne by agents who do not benefit from diversity. We codify this via two desiderata capturing *robustness*. These are *no negative externality* – other agents are not hurt – and *monotonicity* – the agent enforcing the constraint does not see a large increase in value. We show in a formal sense that the Nash Welfare rule that maximizes product of agent values is *uniquely* positioned to be robust when diversity constraints are introduced, while almost all other natural allocation rules fail this criterion. We also show that the guarantees achieved by Nash Welfare are nearly optimal within a widely studied class of allocation rules. We finally perform an empirical simulation on real-world data that models ad allocations to show that this gap between Nash Welfare and other rules persists in the wild.

## 1   Introduction

Allocating heterogeneous divisible items (or resources) among agents with different values over these items is a central problem in economics, with literature dating back many decades [43, 9, 37, 44, 5]. These problems have gained importance in computer science and machine learning due to their applications in, among other things, large-scale online advertising [33, 7, 29, 6, 24, 8], resource allocation in datacenters [28] and sharing economy platforms.

Given their wide applicability, there has been a large body of work on elucidating desirable properties of such allocations. One desirable property is *fairness* or *equity* in the value the allocation provides to different agents. Though there is no universally agreed-upon notion of fairness, one appealing notion [44] defines fair allocations as those that are both Pareto-optimal on the value they provide to the agents as well as *envy-free* [43], meaning that no agent should derive more value from another agent's allocation. A less restrictive notion is the so-called Pigou-Dalton principle [17, 35] or *welfarism*, which states that given any fixed total value of the agents, an allocation should prefer distributing these values so that there is no potential transfer of value from an agent with larger value (the "rich") to one with smaller value (the "poor"). This loosely translates to allocations that optimize

a (weakly) concave social welfare function over agents' values. We call rules that optimize separable, symmetric, concave functions of agent values as *welfarist* allocation rules.[1]

Due to their simplicity of implementation and ease of understanding, most allocation rules implemented in practice are welfarist rules. Indeed, even when allocations are supported by prices, for instance in revenue-maximizing ad auctions, by Myerson's celebrated transformation [36, 22] and its generalizations, these can be viewed as optimizing a linear "virtual welfare" function over allocations [10, 12]. Further, envy-freeness can also be implemented by the welfarist *Nash Welfare* rule [37, 21, 44] that optimizes the product of agents' values, and which we will discuss extensively.

## 1.1 Allocations with Diversity

In this paper, we consider a different desirable facet of such allocations – diversity. The notion of fairness presented above attempts to spread value fairly across agents. Suppose, in addition, the items being allocated were also associated with individuals. For instance, consider the following scenario motivated by online advertising – the items are ad view slots in an advertising ecosystem, each item labeled with the attributes, such as race, age, gender, of the viewing individual on a social media site. The agent represents an advertiser who has different preferences over the viewing individuals. The agent may not only seek an allocation of slots that maximizes its own value, but may also want these slots to not be all views by any particular race. In other words, the agent seeks an allocation that is also diverse on the attributes of the corresponding slots. Similarly, in a team formation platform, an agent seeking such a team may value diversity in attributes such as gender and race of the team members in addition to competence. Diversity has become an increasingly important consideration for eliminating bias in allocation platforms where the items map to humans. Allowing agents to specify some form of diversity constraints on allocations are one way for achieving this goal [45, 4, 11, 38].

The simplest diversity constraint we consider is the case where the agent wants items with different attributes (such as race and gender) in fixed proportions, which models the diversity desiderata in the advertising and team formation settings considered above. We term these "proportionality constraints"; see Section 2.2 for a formal definition. More generally, such constraints capture *complementarity* in the allocation, where an agent can say "I want at least as much of this attribute as that attribute." A different application for such constraints is in machine allocation in data centers, where a client could request machines in different geographic regions in certain proportions [41]. For simplicity, we call these constraints *diversity constraints*, though it should be kept in mind that they model complementarity in general and can easily arise in settings beyond achieving diversity.

In Section 2.2, we model diversity constraints imposed by agent $i$ as simply a convex polytope $\mathcal{P}_i$ such that the empty allocation $\vec{0} \in \mathcal{P}_i$. First, this means such constraints preserve feasibility of the allocation. More importantly, these constraints, including proportionality constraints considered above, can have both positive and negative coefficients, and this leads to interesting and non-obvious behavior of welfarist allocation rules. This forms the focus of this paper.

**Externality in Diverse Allocations.** One natural solution to incorporating diversity is for the allocation platform to ask agents for their constraints and add them to the welfarist optimization problem. Indeed, recent work on large scale online allocation problems [18, 19, 24] shows that even in the presence of *arbitrary* convex constraints on the allocations that can run across time, the resulting problem can be viewed as online stochastic convex programming and admits efficient approximation algorithms [3, 8]. Similarly, recent works on auctions with diversity constraints [12, 14, 26] also follow this route and simply add these constraints to an optimization procedure.

In this paper, our main focus is to understand how adding such constraints affects the *outcome* of the allocation. More precisely, for any welfarist allocation rule, when an agent imposes diversity constraints, this changes the allocation of not just this agent, but all other agents as well. Naturally, seeking a diverse allocation will incur a cost in terms of diminished total value of all the agents. This cost is, of course, counterbalanced by the benefit of diversity to the agent seeking diversity (which does not explicitly appear in our model). Since the agent seeking diversity is the one benefiting from the constraints, it would be desirable from both the other agents' and the platform's perspective that this agent should be the one bearing the majority of this cost, while the values for other agents should

---

[1]We borrow this term from [39], though the concept itself is classic [32, 34] (See Section 2.1 for formal definitions.)

not be significantly impacted. In other words, there should be *little negative externality* on other agents. We present this desideratum formally in Section 2.3.

It is not *a priori* clear whether the welfarist allocation rules are robust in the above sense and thus the extent to which imposing diversity constraints, even by one agent, may impact other agents. In fact, this lack of clarity may be a major reason for a platform to hesitate to implement functionality enabling agents to specify such diversity constraints, a significant shortcoming for applications such as team selection and online advertising. Thus the focus of our work is on quantifying the potential negative externality and on developing recommendations for finding robust allocations.

## 1.2 Our Results

To start with, one could reasonably hope that welfarist allocation rules that attempt to find equitable or fair allocations across agents would be naturally robust to the addition of diversity constraints, in the sense that to a good approximation, the negative externality to other agents' values is bounded. *Our first analytical result shows that the above intuition is false.* In Section 3, we show that *almost all* welfarist allocation rules – regardless of how equitably or fairly they distribute value among agents – are not robust in the sense that the negative externality on other agents that stems from imposing even a single diversity constraint is unbounded (see Theorem 1.) On the positive side, we show that the Nash Welfare objective [37, 21], which optimizes the product of agents' values, is the exception in that it achieves negative externality that is bounded above by an absolute constant factor even when a constant number of agents impose arbitrary diversity constraints. Further, this constant bound is nearly optimal for any number of agents enforcing such constraints since any allocation rule satisfying the Pigou-Dalton principle also suffers similar negative externality. (Theorems 2 and 3.)

We then consider how an agent's imposing diversity constraints affects its own value. We define an allocation rule to be *monotone* if an agent's imposing diversity constraints weakly reduces her own value, and is monotone to a constant factor if the value increases by at most a constant factor. In Section 4, we show such monotonicity is correlated with how egalitarian the allocation rule is – the more egalitarian (or concave) a rule is, the closer to monotone it is (see Theorem 4.) Further, the Nash Welfare allocation rule is monotone to an absolute constant factor.

In Section 5, we use real-world datasets to model ad allocation. For a budget-capped valuation function that is widely studied in this setting, we compare the negative externality induced by various allocation rules. In this setting, the social welfare maximizing rule models first price auctions with budget constrained advertisers, and we empirically show that it indeed suffers large externality and non-monotonicity. However, the Nash Welfare rule suffers very small externality and is monotone, hence making a case for that our theoretical insights will apply in the wild as well.

We summarize the upper and lower bounds on the amount of negative externality and non-monotonicity for various allocation rules in Section 2.3. Taken together, our main contribution is therefore to show that the Nash Welfare objective is uniquely positioned among a large class of welfarist allocation rules to be robust, achieving both no negative externality and monotonicity to within small constant factors. Our results also suggest that care must be exercised by the allocation platform when allowing agents to express diversity or complementarity constraints. If these are directly encoded into the optimization like in [12, 19, 24, 3, 6], they may create second-order unfairness where the resulting decease in value is borne by agents not directly benefiting from the diversity.

## 1.3 Related Work

Our work studies robustness of allocations when users can express diversity constraints. This can be equivalently viewed as an agent specifying a different utility function. For instance, a proportionality constraint corresponds to a user expressing a *Leontief* utility function over the allocation instead of linear utility. In that sense, our results can be equivalently viewed as studying the robustness to agents reporting utility functions that capture diversity. Robustness in our sense has been widely studied when agents can arbitrarily misreport utility functions. In this context, monotonicity is simply strategy-proofness, and non-negative externality maps to non-bossiness [42]. The main result in [42] shows that the only allocation rules that satisfy these properties are variants of serial dictatorships, where agents go one after the other and choose their optimal allocation, and these rules are incompatible with Pareto-optimality and welfarist rules. The only exception is settings resembling matchings [40, 31], which are much more restrictive than the setting we study.

Note however that a diversity constraint is a very specific type of "misreport" – given any allocation, the new utility function is necessarily weakly smaller in value, and this is crucial to all our results. Monotonicity can now be viewed as strategy proofness with such misreports. But now, rules that are not strategyproof in general can easily be monotone. Indeed, we show in Theorem 4 that the MMF rule is actually monotone for general constraints, Pareto-optimal, and welfarist, but is trivially very far from being strategyproof if arbitrary misreports are allowed. In the same vein, the non-negative externality property is much more specific than non-bossiness with general misreports, and it is a priori not obvious that is incompatible with welfarist rules (Theorem 3). One of our main results is to show that even with the specific type of utility "misreports" that diversity constraints imply, this property is incompatible to any factor with welfarist rules, except for the NW rule.

Several recent works have considered the question of computing solutions to discrete optimization problems, such as $b$-matchings [4], stable matching [38], clustering [15], ranking [13, 27], voting [11] and packing [16], when the items belong to groups that must be allocated fairly in the resulting solution. For instance, each cluster in the clustering solution must be assigned a balanced cohort of red and blue points [15]. Our work tackles a normative question that arises in this space – assuming only a subset of agents care about diversity, what do these constraints mean for the allocations to other agents? Extending our work to discrete optimization problems is an interesting open question.

For lack of space, we present a detailed comparison with other notions in Economics and related work on ad auctions in the Supplementary material (Appendix **??**).

## 2 The Diverse Allocation Model

There are $n$ agents and $m$ divisible items, each with unit supply.[2] The goal of the platform is to compute an allocation $\vec{x}$, where $x_{ij}$ represents the fraction of item $j$ that is assigned to agent $i$. Each item is assigned at most once fractionally, that is, $\sum_i x_{ij} \leq 1$ $\forall$ items $j$.

In the context of ad auctions, an item represents a type of viewer or keyword, available in a certain quantity. A fractional allocation would allocate the corresponding fraction of the ad slots of that type to the advertiser (or agent).

We assume that any agent $i$'s value for an allocation $\vec{x_i}$ is linear. We denote the value as $V_i(\vec{x_i}) = \sum_j v_{ij} x_{ij}$, where $v_{ij}$ is the value of the agent for item $j$. Our positive results extend to the more general setting where $V_i$ are arbitrary non-decreasing, continuous, concave functions, while our negative results apply even to the linear case.

### 2.1 Welfarist Allocation Rules

We will consider a general subclass of allocation rules called *welfarist* allocation rules. These optimize $\sum_i f(V_i(\vec{x_i}))$, where $f$ is a monotonically non-decreasing, continuous and concave function.

Formally, given a choice of $f$, the welfarist optimization problem is as follows:

$$\text{Max} \sum_i f(V_i) \quad \text{s.t.} \quad \vec{V_i} \in \left\{ V_i = \sum_j v_{ij} x_{ij} \ \forall i; \ \sum_i x_{ij} \leq 1 \ \forall j; \ x_{ij} \geq 0 \ \forall i, j \right\}$$

Typically, the more concave $f$ is, the more egalitarian across agents is the resulting allocation. The main class of rules we consider are $\gamma$-**fair** rules [34], defined as:

$$f(y) = \frac{y^\gamma}{\gamma}, \qquad \gamma \in (-\infty, 1]$$

At $\gamma = 0$, this rule is defined by its limit as $\gamma \to 0$, so that $f(y) = \ln y$.

As special cases, this yields well-known rules in increasing order of how concave $f$ is, in a sense capturing increasing fairness in the resulting allocations.

**Social Welfare (SW).** $f(y) = y$, corresponding to $\gamma = 1$.
**Nash Welfare (NW).** As $\gamma \to 0$, we obtain $f(y) = \ln y$.

---

[2]The model trivially generalizes to the setting with arbitrary but known supply by scaling the values.

**Max-min Fairness (MMF).** This computes an allocation with $\vec{V}$ such that if $V_i < V_j$, there is no other allocation where agent $i$'s value is larger than $V_i$ and $j$'s value is smaller than $V_j$. This corresponds to $\gamma \to -\infty$.

Among these functions, SW is the least egalitarian, while MMF is the most. We now note some features of welfarist allocation rules that we will use later.

*Pareto-optimality (PO).* The optimal allocation $\vec{x}$ is such that there is no other allocation $\vec{y}$ such that $V_i(\vec{y_i}) \geq V_i(\vec{x_i})$ for all agents $i$, with one inequality strict.

*Pigou-Dalton (PD) Principle [17, 35].* Consider any two allocations with values $\vec{V}$ and $\vec{V}'$ where $\sum_i V_i = \sum_i V_i'$. Fix two agents $k$ and $\ell$ and suppose $V_j = V_j'$ for all $j \neq k, \ell$, while $V_k < V_k' < V_\ell$ and $V_k < V_\ell' < V_\ell$. Then the allocation rule weakly prefers $\vec{V}'$ to $\vec{V}$.

*Anonymity.* We finally say that a rule is *anonymous* if the allocation only depends on the revealed vector of values, and not on the identity of the agents. In particular, agents that reveal identical values receive identical allocations.

Note that all welfarist rules described above satisfy (PO), (PD), and anonymity. The NW rule has additional desirable properties. First, it is *scale invariant*: If an agent $i$ scales all her values $\vec{v_i}$ by any factor $\alpha$ with other agents remaining the same, the optimal allocation does not change. Next, the allocation is *proportional*: Given $n$ agents, the value obtained by any agent in the NW allocation is always at least $1/n$ of the value it could have obtained had it been the only agent in the system.

## 2.2 Specifying Diversity

We assume an agent $i$ can be interested in receiving a diverse mix of items, and can specify diversity as a *constraint set* $\mathcal{P}_i$ on the allocation $\vec{x_i}$ it receives. This constraint set is added to the welfarist optimization problem described above. We assume $\mathcal{P}_i$ is convex and that $\vec{0} \in \mathcal{P}_i$. As mentioned before, such constraints are motivated by settings where the items have attributes related to gender, race, income, etc., and an agent can be interested in obtaining a balanced mix of items, as opposed to a welfare maximizing set of items. We consider two types of constraints:

*Proportionality Constraint.* Here, agent $i$ specifies a partition $S_{i1}, S_{i2}, \ldots, S_{ik_i}$ of a subset $T$ of the items, along with proportions $\alpha_{i1}, \alpha_{i2}, \ldots, \alpha_{ik_i}$ where $\sum_{r=1}^{k_i} \alpha_{ir} \leq 1$. It seeks allocations from each $S_{ir}$ in proportion to $\alpha_{ir}$. In other words, the constraint set $\mathcal{P}_i$ is:

$$\sum_{j \in S_{ir}} x_{ij} = \alpha_{ir} \sum_j x_{ij} \qquad \forall r \in \{1, 2, \ldots k_i\}$$

We can also define an $\epsilon$-approximate notion of proportionality, where the right hand side of the above equality is constrained to be within $(1 \pm \epsilon)$ factor of the left hand side. In our constructions, we will only consider exact proportionality.

*General Constraint.* There is an arbitrary convex set $\mathcal{P}_i$, with $\vec{0} \in \mathcal{P}_i$. The agent needs $\vec{x_i} \in \mathcal{P}_i$. This is more general than the proportionality setting, and could for instance capture multiple proportionality constraints over different partitions of the items. Further, the unconstrained setting where $\mathcal{P}_i$ is all possible allocations is also a special case.

We call an agent who does not specify any constraints an *unconstrained* agent. A constrained agent can specify a general constraint set, and this includes being unconstrained. Our positive results (Theorems 2 and 4) hold in the most general setting with any number of agents who are initially constrained, while the specific agent switches from being unconstrained to expressing a general constraint. Note that this captures as a special case, the setting mentioned before where all agents are initially unconstrained, and one agent switches to expressing a general constraint. Our impossibility results (Theorems 1, 3 and 4, and Corollaries 1 and 3) on the other hand hold even in the special case where there are two agents who are unconstrained and one agent switches to expressing a single proportionality constraint.

## 2.3 Robustness Desiderata for Allocations with Diversity

The high level question we consider is: For welfarist allocation rules, does providing the ability for an agent to express diversity constraints lead to undesirable externality in the resulting allocations?

We study two kinds of externalities: Whether this additional functionality can hurt the agents who do not use this functionality; and whether an agent can gain by misrepresenting themselves as diversity constrained when they are unconstrained in reality. Formally, we consider two natural desiderata for the allocation rule.

*Non-negative Externality.* (NNE) Suppose $\vec{x}$ is the allocation when agent $i$ is unconstrained. Suppose the agent expresses a diversity constraint $\mathcal{P}_i$ and the new allocation with $\mathcal{P}_i$ included in the optimization is $\vec{y}$. Then for any $\ell \neq i$, we should have $V_\ell(\vec{y_\ell}) \geq V_\ell(\vec{x_\ell})$. In other words, if agent $i$ expresses a diversity constraint, it should not create negative externality to other agents. For a parameter $q \in [0, 1]$, we say that an allocation rule is $q$-NNE if $V_\ell(\vec{y_\ell}) \geq qV_\ell(\vec{x_\ell})$ for all $\ell \neq i$. For some of our results, this can be naturally generalized to $k$ agents are initially unconstrained and simultaneously express arbitrary diversity constraints.

*Monotonicity.* (MON) Consider the same setting as above. We should have $V_i(\vec{x_i}) \geq V_i(\vec{y_i})$. In other words, expressing a diversity constraint should not increase the linear value of the agent, since such an increase has the undesirable effect of an agent having the incentive to misreport their constraints. For a parameter $p \in [0, 1]$, we say that an allocation rule is $p$-MON if $V_i(\vec{x_i}) \geq pV_i(\vec{y_i})$.

An allocation rule is $(p, q)$-robust if it is $p$-MON, and $q$-NNE, where $p, q \in (0, 1)$ are constants. As discussed before, robustness is desirable for practical implementation of diversity constraints.

**Summary of Results.** In the sequel, we will bound the $(p, q)$-robustness achievable for various welfarist allocation rules. In particular, we show that among the class of $\gamma$-Fair rules, only the Nash Welfare rule achieves constant $q \geq \frac{1}{4}$, that is, has bounded negative externality. All other rules have $q = 0$. Further, the bound attained by NW cannot be significantly improved: No rule satisfying (PO) and (PD) as defined in Section 2.1 can achieve $q > \frac{1}{2}$.

For monotonicity, we show that $p$ increases with decreasing $\gamma$, that is, as the function $f$ becomes more concave or fair. At one extreme, for MMF, we have $p = 1$, and at the other, for SW, we have $p = 0$. The Nash Welfare objective achieves $p = \frac{1}{2}$, and the bounds we obtain are tight for all $\gamma \in (-\infty, 1]$.

## 3 The Non-negative Externality (NNE) Condition

In this part, we will assume $f$ is differentiable and study which functions $f$ satisfy the $q$-NNE condition for some absolute constant $q > 0$. At first glance, one might intuit that the more concave or fair $f$ is, the more likely it is to satisfy $q$-NNE. Surprisingly, we show this intuition is false. Our main result in this section is that $f$ needs to have a very specific form that is unrelated to its fairness or concavity in order for the allocation rule to be $q$-NNE. In particular, we show that most functions that are not scale-invariant fail the NNE property regardless of how concave they are, while the scale-invariant NW allocation satisfies $q$-NNE for an absolute constant $q$.

### 3.1 An Impossibility Result

Define $g(y) = yf'(y)$. We will assume $g$ is continuous. We need the following technical definition.

**Definition 1.** *For $\delta \leq 1$, we say that $f$ is $\delta$-scaled if $g$ is continuous and $\min_{x,y \geq 0} \frac{g(y)}{g(x)} = \delta$.*

We show an impossibility result for $q$-NNE in the following theorem.[3]

**Theorem 1.** *If $f$ is $\delta$-scaled, then it is not $q$-NNE for any $q > \delta$.*

The above theorem rules out achieving $q$-NNE for any constant $q$ for a wide range functions. In particular, we have:

**Corollary 1.** *The $\gamma$-Fairness rule for any fixed $\gamma \neq 0$ does not satisfy $q$-NNE for any constant $q > 0$.*

This rules out $q$-NNE for any constant $q > 0$ for SW and MMF. A similar corollary holds also for additional functions commonly considered in the literature. For the *Exponential* function $f(y) = 1 - e^{-\lambda y}$, note that $g(y) = \lambda y e^{-\lambda y}$, and $g(0) = 0$. For the *Smooth Nash Welfare* [25, 23] function $f(y) = \ln(1 + y)$, we have $g(y) = \frac{y}{y+1}$, which is increasing with $g(0) = 0$. Similarly, for $f(y) = \ln\ln(1 + y)$, we have $g(y) = \frac{y}{y+1}\frac{1}{\log(y+1)}$ is decreasing with $g(y) \to 0$ as $y \to \infty$. Therefore, these functions are only $\delta$-scaled for $\delta \to 0$, and are hence not $q$-NNE for any $q > 0$.

---

[3]All omitted proofs are in the Supplementary material (Appendix **??**).

## 3.2 Externality of Nash Welfare

We now show quite surprisingly that in a sense, the converse of Theorem 1 is also true. Specifically, this theorem does not rule out achieving $q$-NNE for functions $f$ that are $\delta$-scaled for constant $\delta > 0$. We show that a subclass of these functions satisfy $q$-NNE for constant $q > 0$, and this subclass includes the NW allocation rule. Further, we show that any allocation rule that is Pareto-optimal and satisfies the Pigou-Dalton principle incurs an almost matching negative externality.

**Theorem 2.** *Allocation rules where $g(y) = yf'(y)$ is non-increasing and $f$ is $\delta$-scaled satisfy $q$-NNE for $q = \frac{\delta}{\delta+3}$.*

The next corollary follows by directly applying the above theorem on NW rule. The proof follows by observing that for NW, $f(y) = \ln y$, so that $g(y) = 1$ is a constant, so that it is $\delta$-scaled for $\delta = 1$.

**Corollary 2.** *The NW allocation rule satisfies $q$-NNE for $q = \frac{1}{4}$.*

We note that there could be other functions $f$ satisfying the preconditions of Theorem 2. For instance, $f(x) = 2\ln x - \ln(1+x)$, which is a combination of Nash Welfare and Smooth Nash Welfare, is $\delta$-scaled for $\delta = \frac{1}{2}$, so that it is $q$-NNE for $q = \frac{1}{7}$.

To complete the picture, we now show that no rule that satisfies Pareto-optimality (PO) and the Pigou-Dalton principle (PD) as defined in Section 2.1 can achieve $q$-NNE for $q > 1/2$.

**Theorem 3.** *No allocation rule that satisfies (PO) and (PD) satisfies $q$-NNE for any constant $q > \frac{1}{2}$.*

**Observations and Extensions.** Theorem 2 does not require $V_i(\vec{x_i})$ to be a linear function of $x_i$. It holds as long as $V_i(\vec{x_i})$ is any concave non-decreasing function as long as $V_i(\vec{0}) = 0$. The same holds for all the positive results we subsequently present, e.g., Theorem 4.

As an extension of Theorem 2, suppose $k$ agents express diversity constraints. A naive application yields $q$-NNE for $q = \frac{\delta^k}{(\delta+3)^k}$, since each of the $k$ agents expressing a diversity constraint decreases the value of another agent by a factor of at most $\frac{\delta}{\delta+3}$. However, we show that as long as a small subset of agents expresses diversity constraints, the externality for $\delta$-scaled functions is bounded.

**Corollary 3.** *For any $k$, suppose a set $S$ of $k$ agents switch from being unconstrained to expressing general diversity constraints. Then, allocation rules where $g$ is non-increasing and $f$ is $\delta$-scaled satisfy $q$-NNE for $q = \frac{\delta}{2k+\delta+1}$.*

Therefore, the NW allocation is $q$-NNE for $q = \frac{1}{2(k+1)}$. We show a matching lower bound below.

**Corollary 4.** *There are instances where any allocation satisfying (PO), (PD), and anonymity is not $q$-NNE for $q = \frac{1+\epsilon}{k+1+\epsilon}$, where $\epsilon > 0$ is any constant when each of $k$ agents switches to expressing a proportionality constraint.*

# 4 The Monotonicity (MON) Condition

Unlike the case of NNE where almost all rules other than NW fail, for monotonicity, we show that increasing fairness or concavity of $f$ is correlated with achieving $p$-MON for constant $p > 0$. This is surprising since one would expect that if a diversity constraint hurts another agent by an arbitrary amount, it should help the agent enforcing the constraint also by an arbitrary amount. However, we show that this intuition is false and MON and NNE are indeed very different properties.

For $\gamma$-Fairness, we now show nearly matching upper and lower bounds on $p$-MON as a function of $\gamma$, and show that $p$ decreases with increasing $\gamma$.

**Theorem 4.** *For $\gamma \in (-\infty, 1]$, the $\gamma$-Fairness rule satisfies $p$-MON where $p$ is the solution to $p^{1-\gamma} + p = 1$. Further, this bound on $p$ is tight.*

The above theorem shows that $p$ increases with decreasing $\gamma$, confirming the intuition that achieving $p$-MON is easier as the function becomes more concave. In particular, we have the following corollary by setting $\gamma$ appropriately:

**Corollary 5.** *The following bounds on $p$-MON are tight: $p = 0$ for social welfare ($\gamma = 1$); $p = \frac{1}{2}$ for Nash welfare ($\gamma \to 0$), and $p = 1$ for max-min fairness ($\gamma \to -\infty$).*

Therefore, the MMF allocation rule satisfies monotonicity, combined with (PO) and (PD), while Theorem 3 rules out a corresponding result for 1-NNE.

# 5  Empirical Simulation

We now augment our theoretical analysis with a quantitative empirical demonstration of externality and monotonicity achieved by various allocation rules. We consider a budget-constrained valuation function that is widely used in ad allocations [33, 20, 18]. We generate instances from real-world datasets to mimic bidding data in ad allocation, and empirically compare the externality and monotonicity properties of various allocation rules. We show that even empirically, rules other than Nash Welfare incur significant negative externality, thus providing evidence that our results are not merely a worst-case analysis, but correspond to what may also happen in practice.

**Datasets and Setup.** We use two datasets. The UCI Adult dataset [1] tabulates census information such as geographic area, job, gender, and race for around 50,000 individuals. The Yahoo A3 dataset [2] contains bid information for advertisers for a set of ads.

For the Adult dataset, we consider the two genders as items, and the $14$ job categories as agents. The value of a gender (item) to a job (agent) is set to be linearly and positively correlated with the number of people of that gender working in that job. We take the first $500$ data points. For the Yahoo dataset, we consider the advertisements as items, and the advertisers as agents. The value of an advertisement to an advertiser is set as its bid on the advertisement.[4] We take the first 10,000,000 data points. We arbitrarily take 20 advertisements, together with 6 advertisers who have bid on most of these advertisements. In both cases, the diversity constraint for an agent is the proportionality constraint that equalizes the allocation across items where this agent has non-zero values. (We exclude items with zero value since the agent is clearly not interested in this item.)

In order to make the valuation function mirror advertising applications, we use the budget-capped valuation function [33, 20], where for each agent $i$, its value is

$$V_i(\vec{x_i}) = \min\left(B_i, \sum_j v_{ij} x_{ij}\right). \tag{1}$$

where $B_i$ is the budget of the agent. This function has also been termed *liquid welfare*, since it is an upper bound on the amount of welfare the platform can generate given budget constraints of the advertisers. In our simulation, we draw $B_i$ independently and uniformly from $(0, T)$. For the Adult dataset, we set $T = 20$, and for the Yahoo dataset, we set $T = 10$. These are chosen commensurate with the magnitude of the values of the agents for items.

**Simulation.** We conduct three sets of simulations on each dataset and compare the allocation rules for $\gamma$-Fairness with $\gamma = 1$ (Social Welfare), $\gamma = 0.5$, $\gamma = 0.1$, $\gamma \to 0$ (Nash Welfare), and $\gamma = -1$ (approximating MMF).

In the first *single agent* simulation, a specific agent expresses the diversity constraint, requiring equal allocation of all items with non-zero values. For each allocation rule and each agent $i$, we record the largest $q_i$ such that this rule satisfies $q_i$-NNE when this agent expresses a diversity constraint. We report $q_{\min} = \min_i q_i$ for each rule. In the second *double agent* simulation, two agents simultaneously express diversity constraints, each requiring equal allocations of all items with non-zero value. As before, we compute the minimum $q$ value achieved over all pairs of agents.

For both these simulations, to prevent the negative externality from being dominated by tiny values, when computing the negative externality, we ignore agents whose values in the unconstrained case are less than $10\%$ of their budget.[5] We repeat these simulations 10 times with random seeds.

In the third *monotonicity* simulation, we again consider the case where a single agent expresses a diversity constraint, requiring its allocation of all items to be equal. For each allocation rule and agent $i$, we record the largest $p_i$ such that this rule satisfies $p_i$-MON when this agent expresses a diversity constraint, and report $p_{\min} = \min_i p_i$ for each rule.

---

[4]In reality, the value is the bid times the CTR. Though our dataset does not have CTR information, we believe our results will extend to that case.

[5]Including these preserves performance of NW and $\gamma$-Fairness for $\gamma = 0.1$, while other rules only do worse.

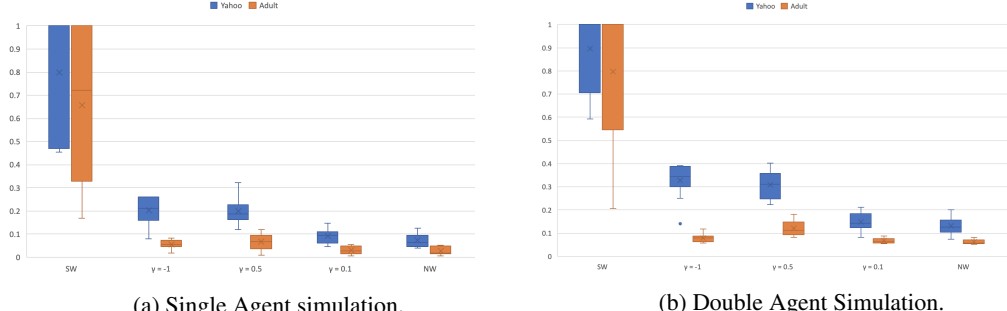

(a) Single Agent simulation.  (b) Double Agent Simulation.

Figure 1: The plots of $(1 - q_{\min})$ for Social Welfare (SW), Nash Welfare (NW), and $\gamma$-Fairness for $\gamma = 0.5, 0.1, -1$.

**Results.** In Figures 1a and 1b, we present the box plots of $(1 - q_{\min})$ for the single and double agent simulations respectively. For both datasets, Nash Welfare and $\gamma$-Fairness for $\gamma = 0.1$ give the best performance with a loss in value below $10\%$ in most cases, while $\gamma = 0.5$, SW, and $\gamma = -1$ are significantly worse, with negative externality at least twice as large. The exception is the good performance of $\gamma = -1$ on the Adult dataset. Moreover, in the double agent simulation, NW and $\gamma$-Fairness for $\gamma = 0.1$ perform much better than the worst-case bounds presented before.

For the monotonicity simulation, we find that $p_{\min} = 1$ for all rules except SW. In other words, these rules are monotone. For SW, we have $p_{\min} = 0.36$ for Yahoo and $p_{\min} = 0.94$ for Adult.[6]

**Relevance to Ad Auctions.** The Social Welfare maximizing rule under the valuation function from Eq (1) closely mirrors allocation rules used in ad auctions [33, 18, 24, 46]. In particular, via standard LP duality, the SW allocation has the following structure: It is a first price auction where bid (or value) of an advertiser is scaled down by an advertiser-dependent parameter. Such an allocation rule mirrors the widely used smooth delivery allocation rules [46] for budget constrained advertisers. Further, rules that allocate adwords in an online fashion essentially compute an approximately SW allocation [33, 18]. Our empirical results show that such a rule suffers large negative externality when advertisers are allowed to express diversity constraints, and this can be mitigated if the platform instead uses Nash Welfare. This complements a recent line of work in advertising on implementing Nash Welfare [8] and regularization [6] as approaches to mitigate unfairness; we show that these approaches mitigate negative impacts of diversity constraints as well.

## 6  Conclusion

The conceptual message of this paper is that incorporating diversity constraints into an allocation platform requires careful selection of the underlying optimization algorithm to prevent negative externality. One could of course wonder whether negative externality is necessarily a bad thing since it forces other agents to consider diversity themselves. However, our results should be viewed as saying that many allocation rules change values of other agents in an unpredictable or counterintuitive fashion. Since the agents typically are not symmetric in either values or what constraints they desire, the resulting externalities will also be asymmetric. This argues for minimizing such negative externality in the first place instead of using it as a disincentive tool.

Our work is just a first step towards understanding the robustness of allocation rules with diversity constraints, and we now present several open questions. In addition to tightening our bounds, it would be interesting to study pricing rules and auctions. Though we do not present details, these allocation rules suffer from similar drawbacks. For instance, truthful auctions [30] or online allocations [33] with budgets are not Pareto-optimal with diversity constraints, while rules that compute optimal auctions [36, 12] or market-clearing solutions [5] are not robust to any approximation even with quasi-linear utilities. We leave a deeper examination of rules with prices and budgets as an interesting open question. Finally, it would be interesting to study such externality in contexts other than allocation problems, for instance, discrete ML problems such as ranking or clustering.

---

[6]The performance of SW is unstable under different sets of random budgets, while the other four rules are stable. Thus, for the sake of exposition, we select a bad instance for SW to present here.

## Acknowledgments and Disclosure of Funding

This work is supported by NSF awards CCF-1637397, CNS-1943584, CNS-1956435, CNS-1916153, CCF-2113798, ONR award N00014-19-1-2268, DARPA award FA8650-18-C-7880, and gifts from Google and Microsoft.

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
