# Supplementary Material

## A   Other Related Work

The Nash Welfare mechanism is classic approach to finding equitable allocations, and achieves fairness via the notion of *market clearing* [5, 9, 42]. It was shown in [23] that this objective finds the solution to the special case of the *Fisher market* [9]. Given linear valuation functions of agents and equal initial budgets, this market computes equilibrium prices for the items, such that when each agent buys their value maximizing allocation subject to exhausting their budget, then each item with positive price is fully allocated. This concept is also called *competitive equilibrium with equal incomes* (CEEI) and is widely studied as a fair allocation rule [35, 11, 49]. The Fisher market with linear valuation functions satisfies the *Gross Substitutes* property [33], which states that if the prices of some items increase, the demand for the other items cannot go down. This implies adding (resp. removing) an agent weakly reduces (resp. increases) the value obtained by the other agents. This property is also called competition monotonicity [36].

Though our results about the externality induced by Nash Welfare appear superficially similar to Gross Substitutes, we cannot find a formal connection. This is because our setting considers the same set of agents but adds constraints, and these constraints do not preserve the Gross Substitutes property. Furthermore, in contrast to the Gross Substitutes property that holds in a strict sense for linear valuation functions, in Section 3, we present a lower bound showing any welfarist allocation rule must result in some negative externality.

Our work considers the pure allocations problem with additional diversity constraints. In some settings such as online advertising, it is also possible to consider allocations with prices and budgets. For instance, if the utility of an agent is its value minus the price it pays (called *quasi-linear*), then the dual of the social welfare maximizing allocation also yields an equilibrium [5, 33]. Similarly, the online algorithms literature [38, 20, 21, 3, 6] considers the model where the total value derived by an agent is constrained by its budget, while auction literature [41, 12, 14, 16, 32] finds allocations and prices where agents do not have incentive to misreport. As mentioned before, though some of these mechanisms suffer from the same drawbacks as welfarist allocation rules, we leave a deeper examination of rules with prices and budgets (from the perspective of robustness) as an interesting open question.

## B   Omitted Proofs

Our proofs will extensively use the following property of the optimal allocation under welfarist rule $f$. Let $\vec{V^*}$ denote the values of the agents in the optimal solution, and let $\vec{V}$ be the values in any other feasible allocation. Since the valuation function is concave and continuous and so is $f$, the space of feasible $\vec{V}$ is convex. This implies the following gradient optimality property:

$$\nabla f(\vec{V^*}) \cdot (\vec{V} - \vec{V^*}) \leq 0 \tag{2}$$

### B.1   Proof of Theorem 1

*Proof.* Consider agents 1 and 2, and items $a$ and $b$. Set $v_{1a} = \alpha$, $v_{2b} = \beta$, and $v_{1b} = v_{2a} = 0$. (We can set the latter values to be any small $\epsilon > 0$ as well.) In the absence of diversity constraint, we clearly have $x_{1a} = x_{2b} = 1$. So $V_2 = \beta$.

Now, agent 1 expresses a diversity constraint and requires $x_{1a} = x_{1b}$. Set $x_{1a} = x_{1b} = x$. Then, we optimize:
$$\max \ell(x) = f(\alpha x) + f(\beta(1-x)), \;\; x \in [0,1].$$

We will show that $\ell'(1-\delta) > 0$. Since $\ell$ is concave, this implies that the optimal solution $\tilde{x} > 1 - \delta$, which in turn implies that $V_2 < \delta\beta$, which thus shows that the allocation rule is not $q$-NNE for $q = \delta$.

To show this, we observe:

$$\ell'(1-\delta) = \alpha f'(\alpha(1-\delta)) - \beta f'(\delta\beta) \geq g(\alpha) - \frac{g(\beta\delta)}{\delta},$$

where the final inequality follows since $f'(\alpha(1-\delta)) < f'(\alpha)$ by the concavity of $f$. We now set $\alpha = \text{argmax}_y g(y)$ and $\beta = \frac{y_\delta}{\delta}$, where $g(y_\delta) = \delta \max_y g(y)$. This yields $\ell'(1-\delta) > 0$, completing the proof. $\square$

## B.2  Proof of Corollary 1

*Proof.* For $\gamma$-Fairness, note that $g(y) = x^\gamma$ and is monotonically increasing and unbounded if $\gamma \in (0,1]$ and is monotonicially decreasing and unbounded when $\gamma < 0$. Therefore, in either case, it is not $\delta$-scaled for any $\delta > 0$, which shows that there is no constant $q > 0$ for which the allocation is $q$-NNE. $\square$

## B.3  Proof of Theorem 2

*Proof.* Suppose agent $i$ imposes a diversity constraint. Let $\vec{A}$ denote the values of the agents in the optimal allocation if the constraint is not enforced, and $\vec{B}$ the vector of values in the allocation if the constraint is enforced. Let $r_k = \frac{A_k}{B_k}$. Consider an agent $\ell \neq i$.

Since $\vec{B}$ corresponds to the values in a feasible allocation even without agent $i$'s constraint, by the gradient optimality condition (Eq (2)), we have:

$$\sum_k \left( g(A_k)\frac{1}{r_k} - g(A_k) \right) = \sum_k f'(A_k)(B_k - A_k) \leq 0,$$

which can be rewritten as:

$$g(A_\ell)\frac{1}{r_\ell} + \sum_{k \neq \ell} g(A_k)\frac{1}{r_k} \leq \sum_k g(A_k). \tag{3}$$

Similarly, suppose we take the allocation without agent $i$'s constraint and remove agent $i$'s allocation from it, the resulting allocation is feasible for the problem where agent $i$ has a constraint. This is because the empty allocation is feasible for agent $i$'s constraints. Applying Eq (2) again, we have:

$$-g(B_i) + \sum_{k \neq i}(g(B_k)r_k - g(B_k)) = f'(B_i)(0 - B_i) + \sum_{k \neq i} f'(B_k)(A_k - B_k) \leq 0,$$

which can be rewritten as:

$$g(B_\ell)r_\ell + \sum_{k \neq i,\ell} g(B_k)r_k \leq \sum_k g(B_k). \tag{4}$$

Since $g$ is non-increasing, so is $g(x)/x$. Therefore, by the Rearrangement inequality, we have for all $k \neq i, \ell$:

$$g(A_k)\frac{1}{r_k} + g(B_k)r_k = \frac{g(A_k)}{A_k}B_k + \frac{g(B_k)}{B_k}A_k \geq \frac{g(A_k)}{A_k}A_k + \frac{g(B_k)}{B_k}B_k = g(A_k) + g(B_k). \tag{5}$$

We now have the following, where the first inequality follows from Eq (5),and the final inequality follows by adding Equations (3) and (4):

$$g(A_\ell)\frac{1}{r_\ell} + g(B_\ell)r_\ell + \sum_{k \neq i,\ell}(g(A_k) + g(B_k))$$

$$\leq \left( g(A_\ell)\frac{1}{r_\ell} + \sum_{k \neq \ell} g(A_k)\frac{1}{r_k} \right) + \left( g(B_\ell)r_\ell + \sum_{k \neq i,\ell} g(B_k)r_k \right)$$

$$\leq \sum_k (g(A_k) + g(B_k)).$$

Simplifying, this further gives:

$$g(A_\ell)\frac{1}{r_\ell} + g(B_\ell)r_\ell \leq g(A_\ell) + g(B_\ell) + g(A_i) + g(B_i).$$

Dividing both sides by $g(B_\ell)$ gives:

$$r_\ell \leq 1 + \frac{g(A_\ell)}{g(B_\ell)} + \frac{g(A_i)}{g(B_\ell)} + \frac{g(B_i)}{g(B_\ell)} \leq 1 + \frac{3}{\delta}.$$

i.e. $\frac{A_\ell}{B_\ell} = r_\ell \leq 1 + \frac{3}{\delta}$. Therefore, $A_\ell \leq B_\ell \left(1 + \frac{3}{\delta}\right)$, showing $q$-NNE for $q \geq \frac{\delta}{\delta+3}$. $\qquad\square$

### B.4 Proof of Theorem 3

*Proof.* Consider the following example: There are 2 items $a, b$ and two agents $1, 2$. The values are $v_{1a} = v_{2a} = v_{2b} = 1$, and the rest of the values are zero. Agent 2 enforces the proportionality constraint $\epsilon x_{2a} = x_{2b}$. First consider the setting with no proportionality constraint. Suppose a rule allocates $x_{1a} = 1 - x$, so that her value is $V_1 = 1 - x$. Then $x_{2a} = x$ and $x_{2b} = 1$ so that the value of agent 2 is $V_2 = 1 + x$. Since the allocation rule satisfies (PD), this forces $x = 0$, so that $V_1 = 1$. Now suppose agent 2 enforces the proportionality constraint and as before, let $x_{2a} = x$. This forces $x_{2b} = \epsilon x$, so that $V_2 = (1 + \epsilon)x$. As before $V_1 = 1 - x$. If $x < \frac{1}{2+\epsilon}$, this allocation cannot satisfy (PD). Therefore $V_1 \leq \frac{1+\epsilon}{2+\epsilon}$ for any allocation satisfying (PO) and (PD). Now taking $\epsilon \to 0$ shows that the allocation cannot be $q$-NNE for any constant $q > \frac{1}{2}$. $\qquad\square$

### B.5 Proof of Corollary 3

*Proof.* Let $\ell \notin S$ denote the agent whose value we are bounding. The inequalities obtained by generalizing Eq (3) and (4) to omit the set $S$ instead of a single agent $i$ now yields:

$$g(A_\ell)\frac{1}{r_\ell} + \sum_{k \neq \ell} g(A_k)\frac{1}{r_k} \leq \sum_k g(A_k), \quad g(B_\ell)r_\ell + \sum_{k \notin S \cup \{\ell\}} g(B_k)r_k \leq \sum_k g(B_k).$$

The same line of reasoning gives:

$$g(A_\ell)\frac{1}{r_\ell} + g(B_\ell)r_\ell \leq \sum_{k \in S \cup \{\ell\}} \left(g(A_k) + g(B_k)\right).$$

Dividing both sides by $g(B_\ell)$ gives:

$$r_\ell \leq 1 + \frac{g(A_\ell)}{g(B_\ell)} + \sum_{k \in S} \left(\frac{g(A_k)}{g(B_\ell)} + \frac{g(B_k)}{g(B_\ell)}\right) \leq 1 + \frac{2k+1}{\delta},$$

thus showing $q$-NNE for $q \geq \frac{\delta}{2k+\delta+1}$. $\qquad\square$

### B.6 Proof of Corollary 4

*Proof.* The lower bound follows by extending Theorem 3. There is an item $a$ such that for agent $1$, $v_{1a} = 1$. For $i \in \{2, 3, \ldots, k+1\}$, there is an item $i$ such that agent $i$ has $v_{ii} = 1$. For $i \in \{2, 3, \ldots, k+1\}$, we also have $v_{ia} = 1$. All other values are zero. Without the diversity constraint, suppose $x_{2a} = x_{3a} = \cdots = x$ and $x_{1a} = 1 - kx$, then $V_2 = V_3 = \cdots = 1 + x$ and $V_1 = 1 - kx$. Then (PD) implies $x = 0$ so that $V_1 = 1$.

Now suppose each agent $i \in \{2, 3, \ldots, k+1\}$ express the proportionality constraint $\epsilon x_{ia} = x_{ii}$. If $x_{1a} = 1 - ky$, then by anonymity, we have $x_{ia} = y$ and $x_{ii} = \epsilon y$ for all $i \in \{2, 3, \ldots, k+1\}$. Therefore, $V_1 = 1 - ky$ and $V_2 = V_3 = \cdots = (1 + \epsilon)y$. Now, any allocation with $y < \frac{1}{k+1+\epsilon}$ does not satisfy (PD). Therefore, any allocation satisfying (PO), anonymity, and (PD) has $V_1 \leq \frac{1+\epsilon}{k+1+\epsilon}$. This completes the proof. $\qquad\square$

### B.7 Proof of Theorem 4

*Proof.* We first present the upper bound. The allocation maximizes $\sum_i \frac{1}{\gamma}V_i^\gamma$. As in the proof of Theorem 2, let $\vec{A}$ denote the vector of values of the agents if agent $i$ did not have a diversity constraint, and let $\vec{B}$ denote the vector of values if the constraint is enforced. Assume by scaling all values by

the same amount that $A_i = 1$, and denote $B_i = x$. Our goal is to upper bound $x$, which will yield the value of $p$. We will assume below that $\gamma \neq 0$, and present the proof for $\gamma \to 0$ separately.

The gradient optimality condition Equation (2) now simplifies to:

$$\sum_i V_i(V_i^*)^{\gamma-1} \leq \sum_i (V_i^*)^\gamma. \tag{6}$$

Consider the unconstrained allocation, but set agent $i$'s allocation to zero. This is feasible for the constrained version whose optimal solution is $\vec{B}$. Applying Eq (6), we have:

$$x^\gamma + \sum_{k \neq i} B_k^\gamma \geq \sum_{k \neq i} \frac{A_k}{B_k^{1-\gamma}}. \tag{7}$$

Similarly, the constrained allocation $\vec{B}$ is feasible for the unconstrained problem. Applying Eq (6), we have:

$$1 + \sum_{k \neq i} A_k^\gamma \geq x + \sum_{k \neq i} \frac{B_k}{A_k^{1-\gamma}}. \tag{8}$$

Combining Equations (8) and (7), we have

$$x - x^\gamma \leq 1 + \sum_{k \neq i} \left( A_k^\gamma + B_k^\gamma - \frac{A_k}{B_k^{1-\gamma}} - \frac{B_k}{A_k^{1-\gamma}} \right). \tag{9}$$

We will now show that

$$A_k^\gamma + B_k^\gamma - \frac{A_k}{B_k^{1-\gamma}} - \frac{B_k}{A_k^{1-\gamma}} \leq 0.$$

This is equivalent to:

$$\left( \frac{1}{A_k^{1-\gamma}} - \frac{1}{B_k^{1-\gamma}} \right) (A_k - B_k) \leq 0.$$

It is easy to check that for any $\gamma \in (-\infty, 1]$, we have $A_k \geq B_k$ iff $A_k^{1-\gamma} \geq B_k^{1-\gamma}$. This proves the inequality. Plugging it into Eq (9), we have

$$x - x^\gamma \leq 1.$$

It is easy to check that for $\gamma \in (-\infty, 1]$, $x$ is maximized when $x = 1 + x^\gamma$, completing the proof.

**Nash Welfare.** When $\gamma \to 0$, the above proof does not directly apply. Nevertheless, we can obtain the bound $p = \frac{1}{2}$ as follows. We follow the same outline as the proof of Theorem 2, but do not fix another agent $\ell$. For NW rule, Eq (3) can be rewritten as:

$$\frac{1}{r_i} + \sum_{k \neq i} \frac{1}{r_k} \leq n.$$

and Eq (4) implies

$$\sum_{k \neq i} r_k \leq n.$$

Our goal now is to upper bound $\frac{1}{r_i}$. Therefore, we solve:

$$\min \sum_{k \neq i} \frac{1}{r_k} \quad \text{s.t.} \quad \sum_{k \neq i} r_k \leq n, \ r_k \geq 0.$$

This implies $r_k = \frac{n}{n-1}$ in the optimal solution, so that $\sum_{k \neq i} \frac{1}{r_k} \geq \frac{(n-1)^2}{n}$. Therefore

$$\frac{1}{r_i} \leq n - \sum_{k \neq i} \frac{1}{r_k} \leq n - \frac{(n-1)^2}{n} \leq 2.$$

Therefore, $A_i \geq \frac{1}{2} B_i$, which shows $p$-MON for $p \geq \frac{1}{2}$.

**Tight Lower Bound.** There are $n+1$ of agents $\{0, 1, 2, \ldots, n\}$ where $n \to \infty$ and let $c = \beta n$, where $\beta \in (0, 1)$ is a constant to be determined later. There are $2n + 1$ items $\{0, 1, 2, \ldots, 2n\}$. We have $v_{00} = c$; for each $i \in \{1, 2, \ldots, n\}$ we have $v_{0i} = v_{ii} = 1$; and $v_{i(i+n)} = c - 1$. All other values are zero. Agent 0 expresses the proportionality constraint $x_{00} = x_{01} = \cdots = x_{0n}$.

We always have $x_{i(i+n)} = 1$ for $i \in \{1, 2, \ldots, n\}$. In the absence of the diversity constraint, we have $x_{00} = 1$. Suppose $x_{01} = x_{02} = \cdots = x_{0n} = x$ and $x_{11} = x_{22} = \cdots = x_{nn} = 1 - x$. The optimal allocation solves

$$\max \ n\frac{(c-x)^\gamma}{\gamma} + \frac{(nx+c)^\gamma}{\gamma} \qquad x \in [0, 1].$$

It is easy to check that $x = 0$ in this solution, so that $V_0 = nx + c = c = \beta n$.

When agent 0 adds the diversity constraint, we have $x_{00} = x_{01} = \cdots = x_{0n} = x$ and $x_{11} = x_{22} = \cdots = x_{nn} = 1 - x$. Again, the optimization problem is:

$$\max \ n\frac{(c-x)^\gamma}{\gamma} + \frac{((n+c)x)^\gamma}{\gamma} \qquad x \in [0, 1].$$

This yields

$$x = \min\left(1, c\frac{(n+c)^{\frac{\gamma}{1-\gamma}}}{n^{\frac{1}{1-\gamma}} + (n+c)^{\frac{\gamma}{1-\gamma}}}\right).$$

We will constrain $c$ so that

$$(c-1)(n+c)^{\frac{\gamma}{1-\gamma}} \geq n^{\frac{1}{1-\gamma}}, \tag{10}$$

so that at the optimal solution, we have $x = 1$ implying $V_0 = (n+c)x = n + c$. This will show $p = \frac{c}{n+c} = \frac{\beta}{1+\beta}$.

The constraint Eq (10) can be written as $(c-1)^{1-\gamma}(n+c)^\gamma \geq n$. Dividing by $n$ and observing that when $\beta > 0$ and $n \to \infty$, $\frac{c-1}{n} \to \beta$, this constraint reduces to:

$$\beta^{1-\gamma}(1+\beta)^\gamma \geq 1.$$

Setting $\theta = \frac{\beta}{1+\beta}$, this implies that this instance is not $p$-MON for $p > \theta$, where $\theta$ is constrained by $\theta^{1-\gamma} + \theta \geq 1$, completing the proof. $\qquad \square$