# OpenReview forum: "Robust Allocations with Diversity Constraints"
_NeurIPS.cc/2021/Conference — NeurIPS 2021 Poster_

### Official Review · Reviewer_gMgx · 2021-07-13

**Rating:** 5
**Confidence:** 4

**Summary:**

This paper proposes a framework for and and analysis of algorithms that allocate divisible resources to a set of agents when the agents can specify both utilities as well as constraints over the items as well. This is motivated by, e.g., computational advertising settings where we want to allocate users to advertisements so we have a reward (revenue of user) as well as constraints over the distribution (e.g., we do not want to show all job adds to men/women). The framework proposed captures popular allocation rules such as Nash and SW-Max, and the ability to specify constraints can be done through either proportionality or general constraints over the allocation set. The known mechanisms are analyzed in terms of their ability to achieve allocations that come within p,q fractions of allocations that do not impose negative externality and/or are not monotonic.

**Ethical Concerns:**

None at this time -- the framework is interesting and allows for the analysis and tradeoff comparisons that could enable more fair/diverse allocations.

**Limitations And Societal Impact:**

The authors do address the limitations of the work adequately. The paper is well positioned and some of the potential downsides of the work are discussed (e.g., strategically using constraint sets).

**Main Review:**

Overall I quite enjoyed the paper. It is well written and the topic is interesting. The framework proposed is nice and the authors are able to incorporate a wide variety of mechanisms in the framework proposed. The problem is well motivated by the computational advertising and recommendation systems settings -- though some more work from the latter could be included, e.g.,

Alex Beutel, Jilin Chen, Tulsee Doshi, Hai Qian, Allison Woodruff, Christine Luu, Pierre Kreitmann, Jonathan Bischof, and Ed H Chi. Putting fairness principles into practice: Challenges, metrics, and improvements. In Proceedings of the 2019 AAAI/ACM Conference on AI, Ethics, and Society, pages 453–459, 2019.

Burke, R., Voida, A., Mattei, N., Sonboli, N. and Eskandanian, F., 2020. Algorithmic fairness, institutional logics, and social choice. In Harvard CRCS Workshop: AI for Social Good.

My main concern with the paper are two fold: (1) it's not clear to me that this is a machine learning paper or has anything to do with learning. In fact, this seems like a more EC type paper given the format and the proofs provided. The (2) second major issue I address more concretely below, but that this paper does not place itself well in the existing literature and I'm not convinced that some of these results are not already known.

If I read this correctly it seems to be that the NNE property is known as Bossiness in the literature and is already fairly well studied. Given the pretty good background on the other issues in this paper I was surprised to see no discussion of bossiness. Perhaps I am mistaken but I think this is a gross oversight of the paper and makes me think that some of these issues may have been handled already? I checked the supplemental material for the extended discussion and found no mention of the concept.

The statement of bossiness from Papai is: "Nonbossiness, a criterion frequently used in the context of strategyproof allocation, ensures that individuals cannot be bossy, that is, change the assignment for others, by reporting different preferences, without changing their own."

This seems exactly to map here to the constrint expression of NNE.  See for example:

Pápai, S., 2000. Strategyproof assignment by hierarchical exchange. Econometrica, 68(6), pp.1403-1433.

SATTERTHWAITE, M. A., AND H. SONNENSCHEIN Ž . 1981 : ‘‘Strategy-Proof Allocation Mechanisms at Differentiable Points,’’ Reiew of Economic Studies, 48, 587-597.

Hosseini, H. and Larson, K., 2019, May. Multiple assignment problems under lexicographic preferences. In Proceedings of the 18th International Conference on Autonomous Agents and MultiAgent Systems (pp. 837-845).

Note that this literature on strategy-proof allocations is highly relevant and needs to be incorporated because, as you say in the paper, the constraints are a kind of second order utility function and so the strategic aspects are the centerpiece of this paper.

I was also curious that metrics from the healthy literature on diverse matching in the non-divisible space was not discussed. It seems that some of the metrics discussed here should be related to those.

Ahmed, F., Dickerson, J.P. and Fuge, M., 2017, August. Diverse weighted bipartite b-matching. In Proceedings of the 26th International Joint Conference on Artificial Intelligence (pp. 35-41).

Chierichetti, F., Kumar, R., Lattanzi, S. and Vassilvtiskii, S., 2019, April. Matroids, matchings, and fairness. In The 22nd International Conference on Artificial Intelligence and Statistics (pp. 2212-2220). PMLR.

Burke, R., Voida, A., Mattei, N., Sonboli, N. and Eskandanian, F., 2020. Algorithmic fairness, institutional logics, and social choice. In Harvard CRCS Workshop: AI for Social Good.

In terms of making the paper better I would suggest the authors add some kind of table of results. By the end it was hard to keep track of the results that were obtained for which rules and how the bounds helped our understanding of the overall setting. E.g., it wasn't clear to me why we needed two constraint formulations and the paper didn't help clarify this -- a table would highlight where the formulations allow for tighter or loser bounds. A concrete example of the difference between the constraints would help too.

Lines 105-109 there is some confusion about value moving up or down I think.

In the formulation of 2.1 it seems that you allow free disposal, but this is not desired right? For then I could always just not assign things and have fair allocations since \vec{0} is in the set? Or perhaps I misread?

The experiments need more information -- averages over runs, the confidence bounds, etc. As it stands I had a very hard time interpreting the results and have somewhat little confidence in them since I don't know any error measures.

Also for the experiments, it would help to implement the constraints as optimization criteria to check as a baseline. I didn't see where this was done and given the size of the allocations being used a simple LP formulation should be able to handle this. As it stands the results are hard to interpret without a baseline.

-----
Updated after response and discussion.

I have raised my score slightly but in my opinion this paper is still below the threshold. The author response that bossiness is not directly implied I agree with, but not engaging with that literature and not clearly articulating why it is related and what the improvements for this paper are needs to be addressed. These areas are extremely closely related and the fact that you address truthfulness (hence a component of strategyproofness) and don't address bossiness is a shortcoming. I am not saying you should cite all the literature but the paper should cite at least one paper on bossiness since the paper is directly addressing the same question: how bad can a report of one agent affect the allocation to other agents.

**Time Spent Reviewing:**

1.5

---

> ### Author Response · Authors · 2021-08-06
> **Response to Review 4**
>
> We thank the reviewer for carefully reading the paper, and pointing out missing references. We will include the recent works on diverse allocations pointed out.
>
> Comparison with Strategy Proofness: The main complaint of the reviewer is that we did not cite related work on strategy proof allocations, particularly work on “non-bossy” allocations. The main result from this line of work is that the only allocation rules that are strategy proof and non-bossy are serial dictatorships, which are in general far from being Pareto-optimal. There are better results known for more specialized settings than the one we study, for instance Pareto-optimality when an agent can be allocated at most one item. We will present a detailed comparison in the final version, and not including this is an oversight on our part.
>
> That being said, we emphasize that our work does not study strategy-proofness, so the works referenced by the reviewer are only tangentially related to our paper. To see this, let’s adopt the viewpoint of the reviewer and pretend agents are misreporting utility functions. However, a diversity constraint is a very specific type of “misreport” -- given any allocation, the new utility function considers the maximal portion of the allocation satisfying the constraints, and uses the original utility function on this portion. This means the agent “misreports” a utility function that is always smaller in value, and this is crucial to all our proofs. Monotonicity can now be viewed as strategy proofness with such misreports. But now, rules that are not strategyproof in general can easily be monotone, and as we point out to another reviewer’s comment, the welfarist MMF rule is actually monotone, (PO) and (PD)  for general constraints (but is trivially very far from being strategyproof if arbitrary misreports are allowed).
>
> In the same vein, NNE is a much more specific property than non-bossiness with general misreports, and it is a priori not obvious that is incompatible with welfarist rules that satisfy (PO) and (PD) (Theorem 3). One of our main results is to show that even with the specific type of utility “misreports” that diversity constraints imply, NNE is still incompatible to any factor with welfarist rules, except for the NW rule.
>
> Our results are therefore not implied by prior work on strategy proofness in any obvious way, and it is unclear why it would even have been studied under that umbrella. On the contrary, the fact that non-bossiness has been extensively studied for general misreports makes a stronger case for the NNE property as the natural one to consider for our setting. Further, it brings up the interesting open question of generalizing our approximation results to general misreports. We will include this discussion in the final version, and thank the reviewer for pointing this out.
>
> Free disposal: We do allow free disposal; however, we restrict our attention to rules that are Pareto-optimal, which precludes trivial solutions. We do discuss this in the paper, but will highlight it better.
>
> Experiments:  We plotted the worst-case p and q over the agents over a random choice of the budgets. We feel this is appropriate given that such instances can easily arise in an ad platform. We will include a box plot of all values of p,q for completeness.

---

### Official Review · Reviewer_PQ7X · 2021-07-16

**Rating:** 8
**Confidence:** 4

**Summary:**

The authors study the problem of allocation in the presence of a diversity constraint. Any participant can impose a diversity constraint - demanding proportional amounts of multiple items. An allocation can be computed using a social objective such as social welfare, Nash
 welfare subject to this constraint. There are two things that can go "wrong": a participant other than the one imposing the constraint might obtain lower value and the participant imposing the constraint might obtain higher value. The authors ask the question to what extent does the choice of objective guarantee that things won't go "too wrong". That is, any participant not imposing constraint gets at least q fraction of its value and the participant imposing the constraint does not gain more than 1/p factor.

The authors show that for almost any objective that transforms individual participants values and sums them, it is not possible to obtain q = 1. In particular, for social welfare, \gamma-fair, max-min fair objectives it is not possible to get q > 0. In contrast, for Nash Social welfare, one can get q = 1/4. Finally, any rule satisfing pareto-optimality, Pigou-Dalton principle cannot get q > 1/2. The authors also extend these to settings where multiple participants impose constraints.

For the monotonicity - ensuring the constraining participant does not gain too much value - the authors show results for Nash-welfare and \gamma-fair welfare functions. For Nash welfare, 3/4 > p >= 1/2 and for \gamma-fairness, p >= (1-\gamma)^1/\gamma. They also provide an upper bound on p for  \gamma-fairness that is given implicitly by an equation. The \gamma-fairness result implies that for social-welfare p cannot be > 0.

The authors also provide an empirical evaluation of q and p values obtained on real datasets.



**Limitations And Societal Impact:**

Authors have mentioned the limitations of their approach and justified them. The justifications aren't fully satisfactory but the authors approach is still interesting.

**Main Review:**

The model studied by the authors is interesting and the questions the authors explore seem new. The authors do impose a lot of constraints - pareto-optimal, pigou-dalton allocation rule subject to proportionality constraints, and measure its performance against negative-externality, monotonicity criteria. The allocation rules are natural and to some extent a natural implementation when introducing diversity constraints would be to maintain the old allocation rules and impose extra constraints - and this work points out that this can result in non-monotonicity and negative externality and this can be very bad if using social welfare as opposed to Nash welfare as the objective.

The paper is very well written. The presentation could be improved a little by having some proofs or examples in the main body to help build intuition.

The results authors provide are interesting and provide some understanding of the effect of imposing diversity constraints. There are still some aspects that are not fully explored. One question is about monotonicity - the authors do not provide as thorough an analysis of this. In particular it's not discussed if some PO, PD rule could obtain better than 1/2 approximation. Authors also do not provide bounds for the min-max objective function. For the negative externality, Nash Welfare has \delta = 1, so potentially a tighter analysis can provide a better bound better than \delta/\delta + 3, which can match the lower bound of 1/2 for any PO, PD rule.  The examples used in Theorem 3 and Corollary 4 is very skewed - the buyer prefers \epsilon fraction for every unit of another item even though both are valued the same. I am curious if less skewed examples can be constructed perhaps by setting proportionality constraint to be equal but allowing the values to be different.

There are some meta-critiques to the authors approach, the authors have anticipated and discussed these.
* Authors could have found allocations with proportionality, non-monotonicity, no negative externality constraints and assessed how these perform on the social objectives. Authors do mention this and justify their approach for the unique challenges it poses (optimizing for pareto-optimality and proportionality simultaneously is hard.) Their justification is not fully convincing, but I think what they do is reasonable given what I wrote above.
* Authors also consider the effect of introducing proportionality constraint in the ads setting - the proportionality constraint is fairly natural in the ad setting, however the value of the participants might need to be evaluated as quasi-linear utility as opposed to just the value of the allocation. The analysis also becomes more involved if the auction is not truthful. Authors discuss this critique and leave it as open question.

Overall, the results in this paper are interesting and detailed enough to warrant acceptance. The paper will become stronger if some of the bounds were improved.

One minor note, the proof of theorem 1 is buggy. In line 584, f'(\alpha(1-\delta)) > f'(\alpha). It's not clear what y_\delta is also g(y_\delta)/max_y(g(y)) > \delta - this last inequality in particular breaks the proof and some tweak might be required to make it work. Please specify how this proof can be fixed. I did skim through a few other proofs and they seemed correct and thorough.








**Time Spent Reviewing:**

2

---

> ### Author Response · Authors · 2021-08-06
> **Response to Review 3**
>
> We thank the reviewer for carefully reading the paper. We will incorporate the suggestions in the final version.
>
> Analysis of Monotonicity:  Thanks for pointing this out. Our results are indeed more general, and we will add additional exposition to the paper if accepted. In particular, the class of $\gamma$-fair policies can be easily extended to functions of the form $f(x) = \frac{1}{\gamma} x^{\gamma}$ for $-\infty < \gamma \le 1$, that is, allowing negative $\gamma$. This set of functions is still concave and increasing. When $\gamma \rightarrow 0$, this becomes NW, when $\gamma = 1$, this is SW, and when $\gamma \rightarrow - \infty$, this is MMF, thereby including these rules as special cases. We can now strengthen the analysis of Theorem 5 using a similar argument in the current proof to show the following: Consider $\gamma$-Fairness for $\gamma \in (-\infty,1]$. This satisfies $p$-MON for $p$ the solution to $p^{1-\gamma} + p = 1$, which combined with Theorem 6 shows that this value of $p$ is tight for monotonicity. In particular, this will show that MMF is $1$-MON, NW is $½$-MON, and SW is not $p$-MON for any $p > 0$, and the bounds for $\gamma$-Fair functions are tight.
>
> Other Questions: To answer the reviewer’s question about the existence of a PO, PD, and monotone rule, the MMF rule, viewed as the above limit, will satisfy this.
>
> Bug in the proof of Theorem 1:  This is just a typo. We are assuming $y_{\delta} = argmin_y g(y) $,  so that  $ \delta \max_yg(y) = g(y_{\delta}) $. This suffices to fix the proof.

---

### Official Review · Reviewer_NZRm · 2021-07-17

**Rating:** 6
**Confidence:** 4

**Summary:**

The paper considers the problem of allocating unit divisible items under several welfare objectives (i.e., social welfare, gamma-fairness, nash welfare, and max-min fairness) subject to general diversity constraints (modeling as a polytope for each agent). The paper presents two notions of measures, non-negative externality (NNE) and monotonicity (MON), that quantify the impact of the constraints and define their approximated notions of q-NNE and p-MON, respectively.

In particular, they provide a lower bound of q for a class of continuous functions and upper bounds of q for specific welfare functions that depend on the \delta-scaled. For the MON condition, they provide lower and upper bounds for Nash welfare and \gamma-fairness objectives.

Finally, they conduct experiments on two real-world ad allocation datasets (i.e., UCI Adult and Yahoo A3) and using the budget-capped valuation function (along with other parameter settings) to demonstrate the q and p values of various social welfare functions.

**Limitations And Societal Impact:**

Overall, I find the paper very interesting and enjoyable to read. While the results are a bit simple, it is interesting to see the upper and bound lowers of the q-NNE and p-MON for various welfare functions.

I do have some concerns regarding the definition of NNE that considers only one agent (and a set of experiments consider two agents with diversity constraints), the (p, q)-robust that isn't really clear studied (what are the best combinations of two instead of fixing one of them), and the results that for MON seem a bit less general than the MME results. There is also a question about where were the proportional constraints used in the study and some random results on truthfulness.

I also wish the experiments would contain other types of constraints.

All in all, I am slightly positive about this paper (mainly due to its readability, new idea, and its potential insights into the impact of diversity constraints).

**Main Review:**

Abstract:

"to introduce diversity constraints" -> does the planner knows about this? How is this different from the setting where you have multiple diversity constraints (say as input)?
It would also be good to have some examples of the constraints

"demographic parity" -> which is? how would this apply to user ad slots? are you talking about the slots that have race/gender?

"expresses diversity constraints" -> how would the agent express this?

"other agents hurt significantly?" -> there is a need a way to measure hurts

"the cost of introducing" -> how are you defining the cost here?

"We codify this via two desiderata capturing robustness." -> there should be some connections to your previous sentences to robustness in your context (i.e., what are you robust to?)

"does not see a large increase in value" -> w.r.t to?

"the Nash Welfare rule " -> is that an objective here? rather than rule? what is a rule in your context? or allocation?

"when diversity constraints " -> does this depend on the types of the constraints?

"rules fail this criterion" -> when you say criterion, do you mean one of the two properties in the previous sentences?

"class of allocation rules." -> what is the benefit of these guarantees?


"We finally perform an empirical simulation" -> the gap of? it would be good to say a line about the observations

1 Introduction

"competing agents " -> is the competing the right situation? how are they competing? heterogeneous agents?

"their applications in" -> would be good to specific some concrete examples

"Pareto-optimal" -> are you going to shortly define this? i think you define envy-free briefly but not this

"given any fixed total value of the agents" -> is it the sum?
"is no potential transfer of value" -> to achieve what property from the potential transfer?

"separable, symmetric, concave functions" -> it would be good to provide definitions of them more completely if its necessary/important for the paper

"envy-freeness can also be implemented by the welfarist Nash Welfare" -> this can be hard to see ... would Nash welfare implies envy-freeness?

1.1 Allocations with Diversity

"with people" -> individuals

"ad view slots " -> each slot is labeled with race?

"to not be all views by say, white males" -> I would avoid pointing out a specific race

"diverse on the attributes of the corresponding individuals" -> I guess over the slots?

For the provide examples, it would be good to provide some citations for them

"in fixed proportions" -> add (e.g., )

"I want at least as much of this item" -> is this over the attributes or attributes?

"a convex polytope" -> how are you defining the polytope? a set of linear inequalities?

"arbitrary convex long-term constraints" -> can you elaborate?

"one bearing the majority of this cost," -> how will the cost be transferred in such a situation?

1.2 Our Results

"negative externality to other agents’ values is bounded" -> one question I have is that can they always satisfy the diversity constraints? If it doesn't, how can you define the overall values?

" this constant bound is nearly optimal" -> in what sense? is this the best you can do (i.e., a lower bound)?

"reduces its own" -> his/her?

"budget-capped valuation function" -> which is?

"externality and monotonicity to within small constant factors" -> one is applying to the global objective and the other is applying to the agent's utility

"If these are directly encoded into the optimization like " -> it would also be good to show this in experiments

"Pareto-optimal or even proportional " -> how important are these notions if you were to valid them a little bit?

I think it is important to have an explicitly related work section in the paper (since Appendix won't be appearing in the proceedings); perhaps cutting out a bit more our results and 1.1 to fit in the related work

2 The Diverse Allocation Model

2.1 Welfarist Allocation Rules

"used allocation rules below in increasing order " -> could you just measure this by taking a derivative?

is there a citation for MMF or a more precise definition using max min? the current one doesn't seem that precise per say

"We now note some" -> it would be good to say they are useful later on

Is there a more intuitive meaning for PD?

2.2 Specifying Diversity

Can you justify the proportionality constraint a bit more? this seems to be conditional on the items that the agents receive in general (not out of the total allocations)

"factor of the left hand side" -> multiplicative?

What other constraints do you consider beyond proportionality for fairness?

2.3 Robustness Desiderata for Allocations with Diversity

"Non-negative Externality" -> are you only defining this w.r.t. one agent? what if more than one agent change? but also, why would it increase other agents if both agents express diversity constraints?

3 The Non-negative Externality (NNE) Condition

I do have a concern where what if they are impossible to satisfy all the Pi's (no allocation)?

3.1 An Impossibility Result

Definition 1 -> what if g(x) = 0 or g(y) = 0? or greater than 1?

3.2 Externality of Nash Welfare

there seems to be a gap between Theorem 1 and Theorem 2

It's kinda strange that you have a truthful result embedded when the paper is not really about it; i would suggest moving it or discuss it in the conclusion or discussion

could you also comment on the computational aspects of the problems?

Computational aspects of the problems??

4 The Monotonicity (MON) Condition

Do you have any general results (similar to Theorem 2 for more general function)?

what about something like social welfare? max-min fairness?

5 Empirical Simulation

Can you discuss what properties do the budget-capped function satisfy?

"when this agent expresses a diversity constraint" -> did you try other types of diversity constraints? How do the results change?

"In the second double agent simulation" -> here it doesn't seem like your definition covers this case at all?

"In the third monotonicity simulation" -> why can't you measure the same thing for the first and second simulations?

please increase the fonts of the figures, they are hard to see without zooming in

It would be good to see the exact parameters for reproducibility

**Time Spent Reviewing:**

6

---

> ### Author Response · Authors · 2021-08-06
> **Response to Reviewer 2**
>
> We thank the reviewer for a detailed read of the paper. We will address all comments in the final version, and focus below on the important questions raised.
>
> "envy-freeness can also be implemented by the welfarist Nash Welfare"
>
> A: We will put in a citation to [42].
>
> "arbitrary convex long-term constraints" -> can you elaborate?
>
> A: In an advertising scenario, the allocation is performed over time, so these constraints run over time. For instance, by the end of the day, the constraint should hold over the allocation done during the day. This is just an application, so we do not model this aspect.
>
> "one bearing the majority of this cost,"
>
> A: This can be seen in the bad example for Theorem 1 in Section 3.1.
>
> What if they are impossible to satisfy all the Pi's (no allocation)?
> "negative externality to other agents’ values is bounded" -> one question I have is that can they always satisfy the diversity constraints?
>
> A: We assume the zero allocation satisfies the constraints. It’s not hard to see that the proportionality constraint satisfies this property.
>
> "Pareto-optimal or even proportional " -> how important are these notions?
>
> A: Pareto-optimality is a weak, minimal notion of efficiency, while proportionality is a minimal notion of fairness. These are widely studied and implemented.
>
> Is there a citation for MMF or a more precise definition?
>
> A: Please see response to Reviewer 1 and 3. We will expand the definition of $\gamma$-Fairness to negative $\gamma$, and this will capture MMF as a special case.
>
> Is there a more intuitive meaning for PD?
>
> A: Given any fixed value of the agents, allocation should prefer distributing these values so that there is no potential transfer of value from an agent with larger value (the “rich”) to one with smaller value (the “poor”). This interpretation is provided in the beginning of the introduction.
>
> What other constraints do you consider beyond proportionality for fairness?
>
> A: Our results also hold for “packing” constraints, for instance, upper bounds on allocation for specific demographics.
>
> "Non-negative Externality" -> are you only defining this w.r.t. one agent? What if more than one agent changes? but also, why would it increase other agents if both agents express diversity constraints?
>
> A: We do consider the generalization where $k$ agents express these constraints. The increase/decrease could be unpredictable simply because the diversity constraints are arbitrary.
>
> Definition 1 -> what if g(x) = 0 or g(y) = 0? or greater than 1?
>
> A: We have $\delta = min_{x, y} g(y) / g(x) = min_y g(y) / max_x g(x)$. So $\delta  \le 1$. It doesn’t matter if $\min_y g(y) = 0$. However, if $\max_x g(x) = 0$, then we have $g(x) = 0$ for all $x$, which implies that $f’(x) = 0$ for all $x$. This will imply that $f(x)$ is a constant function, thus not concave and therefore not a valid valuation function.
>
> Could you also comment on the computational aspects of the problems?
>
> A: The rules are standard convex optimization problems and could be solved in polynomial time.
>
> 4 The Monotonicity (MON) Condition
> Do you have any general results (similar to Theorem 2 for more general function)? What about something like social welfare? max-min fairness?
>
> A: One could derive this, but since NW is the most interesting case, we chose to present it for NW instead. Please also see rebuttal for Reviewers 1 and 3, where we have outlined a generalization of $\gamma$-Fairness and tightened the result in Theorem 5. Theorem 5 applies to SW by setting $\gamma = 1$. The generalization will capture MMF.
>
> 5 Empirical Simulation
> "when this agent expresses a diversity constraint" -> did you try other types of diversity constraints? How do the results change?
>
> A: We didn’t try other types of diversity constraints, as the proportionality constraint is already fairly general. We believe that the results would be similar for other types of diversity constraints.
>
> "In the second double agent simulation" -> here it doesn't seem like your definition covers this case at all?
>
> A: We do mention the possibility of multiple agents expressing diversity constraints. We will provide a formal definition.
>
> "In the third monotonicity simulation" -> why can't you measure the same thing for the first and second simulations?
>
> A:  The monotonicity simulation is exactly the same as the first externality simulation. They are just measuring different things. We will rephrase it.
>
> It would be good to see the exact parameters for reproducibility
>
> A: All parameters that produce our results are given in the code we submitted. As a side note, all these parameters are randomly generated, which shows the generalisability of our simulation results.

---

### Official Review · Reviewer_xP58 · 2021-07-18

**Rating:** 6
**Confidence:** 3

**Summary:**

The paper studies resource allocation among a group of agents, where
  the agents can express diversity constraints on the allocations they
  receive. Here, the diversity constraints require the allocation to
  satisfy some proportionality constraints on the amount of each
  resource received by the agent, or more generally requires the
  allocation to lie in an arbitrary convex set.  In this setting, the
  authors seek to identify allocation rules that satisfy two natural
  requirements. First, it is desired that the allocation rule satisfy
  non-negative externality, wherein the utility of any agent should
  not decrease due to some other agents' expression of diversity
  constraints. Second, to maintain the right incentives, it is desired
  that the allocation rule satisfy monotonicity, wherein an agent
  cannot increase her utility by expression a diversity
  constraint. The authors study the class of welfarist allocation
  rules, which maximize the sum of a concave, continuous function of
  the agents' utilities. Among the rules in this class, the authors
  show that the Nash welfare rule uniquely satisfies (approximate
  versions of) both the requirements. The authors also show that the
  guarantees of the Nash welfare rule are near optimal, and support
  their theoretical results with numerical simulation.

**Limitations And Societal Impact:**

Yes

**Main Review:**

* Pros
   - The paper studies an important problem of incorporating diversity
     constraints into resource allocation that arises naturally in the
     context of ad auctions or assigning workers to tasks, among
     others.
   - The two requirements on the allocation rules studied in the paper
     are natural, and are driven by economic concerns. Furthermore,
     the characterization of the Nash welfare rule as uniquely
     satisfying (approximate versions) of the two requirements adds
     further justification for the Nash welfare rule, beyond the many
     theoretical properties already known.
   - The authors further consider other common welfarist rules, and
     characterize their failure to satisfy the two requirements.

* Comments
   - It is unclear why the Max-min fairness fits into the class of
     welfarist rules. I am hardpressed to see the rule as maximizing
     the sum of concave functions of the agents utilities.
   - The presentation of the results can be improved. Currently, the
     paper splits the analysis into different sections (corresponding
     to the two requirements). This makes it harder to compare the
     performance of the different rules. Furthermore, the monotonicity
     properties of the SW and the MMF policies are not discussed.


The results in the paper are not sufficient to conclude whether the
Nash welfare rule is the best possible welfarist rule for the two
specified requirements. The upper bounds provided for non-negative
externality ($q=1/2$) are substantially larger than that for the NW
rule ($q=1/4$). For the monotonicity requirement, no bounds on the
performance of a welfarist rule are provided. Nevertheless, I find the
paper makes an interesting theoretical contribution, exhibiting the
good properties of the Nash welfare rule.

**Time Spent Reviewing:**

5

---

> ### Author Response · Authors · 2021-08-06
> **Response to Reviewer 1**
>
> We thank the reviewer (and other reviewers) for pointing out the disconnect between MMF and the rest of the analysis. Our results are indeed more general, and we will add additional exposition to the paper if accepted. In particular, the class of $\gamma$-fair policies can be easily extended to functions of the form $f(x) = \frac{1}{\gamma} x^{\gamma}$ for $-\infty < \gamma \le 1$, that is, allowing negative $\gamma$.  This set of functions is still concave and increasing. When $\gamma \rightarrow 0$, this becomes NW, when $\gamma = 1$, this is SW, and when $\gamma \rightarrow - \infty$, this is MMF, thereby including these rules as special cases. We can now strengthen the analysis of Theorem 5 using a similar argument in the current proof to show the following: Consider $\gamma$-Fairness for $\gamma \in (-\infty,1]$. This satisfies $p$-MON for $p$ the solution to $p^{1-\gamma} + p = 1$, which combined with Theorem 6 shows that this value of $p$ is tight for monotonicity. In particular, this will show that MMF is $1$-MON, NW is $½$-MON, and SW is not $p$-MON for any $p > 0$, and these bounds are tight for $\gamma$-Fairness.
>
> “are not sufficient to conclude whether the Nash welfare rule is the best possible”: We only use the phrase “best possible” in the sense that it has both bounded p and q bounded away from zero, while other rules fail to have non-zero p. We will explain this better.
>
> We will also address the readability of the paper taking into account the suggestions of the reviewer.

---

### Decision · Program_Chairs · 2021-09-27

**Decision:**

Accept (Poster)

**Comment:**

   Overall the reviewers are quite positive about this paper: every reviewer thought the paper studies an interesting topic, and the results are interesting.  A weakness of the paper is that it does not quite fully explore the space. This makes it hard to really compare the different allocation rules, since various subsets of results are shown for each rule. The paper would be substantially strengthened if it were able to give a more complete accounting of the properties of each allocation rule. It is suggest that the authors partially address this shortcoming by giving some sort of overview of the results that can better facilitate comparison.  One reviewer brought up a deficiency in comparing to the existing concept of "bossiness," which seems highly related to the proposed methods. The authors are strongly encouraged to add some comparison to this literature.